# Balance Your Move: Investigating Text to Motion from a Signal Density Measuring View

## Abstract

Multimodal conditional motion generation enables the creation of precise and diverse human motions by combining complementary control signals such as text descriptions and trajectory hints. However, existing methods often rely on static or simplistic fusion strategies, overlooking the fact that the semantic and spatial information of the inputs varies over time. This variation can lead to modality conflicts in which one signal dominates, causing trajectory deviations or semantic drift. Our key insight is that the information density of textual and trajectory signals can serve as a reliable indicator for dynamically balancing their influence during motion generation. Building on this insight, we propose the Signal-Balanced Motion Generator (SBMG), which dynamically measures and leverages the temporal variation of information density to adaptively regulate the relative importance of textual and trajectory signals throughout generation. Experiments on benchmark datasets demonstrate that SBMG significantly enhances both semantic alignment and motion control accuracy, reducing FID by 60.5% and trajectory error by 4.3%, thereby achieving substantial improvements over prior methods in dynamic multimodal motion generation.

## 1 Introduction

Human pose generation is essential for a variety of applications such as virtual character animation, human–computer interaction, virtual reality and robotic motion planning. It has become one of the most active research topics in computer vision. With the growing demand for realistic and controllable digital humans and embodied agents, generating natural and coherent human motion sequences that accurately follow user intentions has attracted increasing attention from both academia and industry.

Recently, diffusion model-based motion generation approaches (Xie et al., 2023; Pinyoanuntapong et al., 2024a; Wan et al., 2024) have attracted increasing attention. These approaches typically take textual descriptions as the sole input and iteratively denoise a motion sequence to satisfy the semantic constraints. To achieve finer control, some works introduce additional guidance signals during the generation process, such as motion trajectories of specific human body keypoints. It is important to note that although the input modality remains text-only, the generation process fuses textual semantics and trajectory signals, which respectively ensure semantic guidance and detailed motion constraints, thereby improving both interpretability and precision.

In these approaches, the explicit input to the model is text only, while an additional trajectory signal is introduced during the generation process as auxiliary guidance. The text provides high-level semantic constraints throughout the denoising steps, whereas the trajectory focuses on controlling fine-grained spatial movements. In this setting, a key challenge is to dynamically balance the influence of the two guidance signals during motion generation, as the model must simultaneously handle information from both the textual and trajectory signals.

Existing works (Shafir et al., 2023; Wan et al., 2024; Xie et al., 2023) typically adopt straightforward fusion or weighting strategies during the generation process to combine two types of guidance signals: semantic constraints from text and spatial constraints from the trajectory. For instance, PriorMDM integrates these signals by concatenating their features, while OmniControl employs static weights to blend them smoothly during generation. However, these methods often assume that both

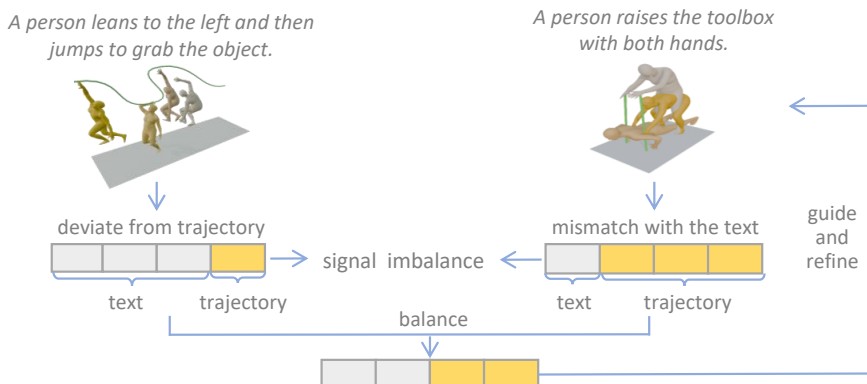

Figure 1: Issues with existing methods include deviation from the intended trajectory and mismatch with the text. Our approach dynamically adjusts the generated motion by balancing guidance from text and trajectory signals.

guidance signals remain equally reliable throughout the sequence, which is rarely true in practice, especially when their information density varies over time or even conflicts.

In diffusion-based motion generation, the text signal consistently serves as the primary driver for denoising, while the trajectory signal plays a stronger constraining role at specific steps. When the text semantics contains compound action instructions and fine-grained trajectory control is injected at certain points in the process, the model may encounter competing directives at the same time step, leading to motion inconsistency or semantic misalignment. For example, in the middle of diffusion generation, if the text instructs "walk forward and then sit down" but the trajectory still corresponds to the walking phase, the model may prematurely generate the sitting motion, resulting in unnatural action transitions. Our analysis shows that in semantically dense periods, the model tends to overemphasize text guidance while neglecting the trajectory, whereas in semantically sparse periods, the reverse occurs — over-reliance on the trajectory reduces semantic expressiveness.

To address these issues, we propose a simple yet effective Signal-Balanced Motion Generator (SBMG). As illustrated in Figure 1, our approach dynamically adjusts the relative influence of two guidance signals during the generation process, namely the semantic guidance from text and the spatial constraints from trajectory, based on the temporal variation of their information density. Specifically, we first compute the guidance strength of each modality by matching control signals with motion features at each time step, enabling us to estimate which modality primarily influences action generation at different points in time. Then, conditioned on the temporal distribution of information density, the model generates a set of dynamic weights for each modality, facilitating cross-modal semantic alignment and harmonization over time. Ultimately, SBMG effectively mitigates action inconsistency and semantic distortion caused by modality conflicts.

We conduct extensive experiments on two benchmark datasets and across three backbone models to validate the effectiveness and generalizability of our method. Results demonstrate that our approach substantially reduces trajectory deviation and semantic misalignment. Notably, on the HumanML3D dataset, our method achieves a 60.5% reduction in FID and a 4.3% decrease in trajectory error, outperforming existing approaches across multiple metrics.

Our main contributions are summarized as follows:

- We provide a systematic analysis of how signal density impacts control effectiveness in multimodal motion planning, and propose a dynamic weighting mechanism to adjust the importance of each modality over time.

- We introduce SBMG, a signal-balanced motion generator that models information density to dynamically regulate the influence of textual and trajectory signals, effectively alleviating issues such as action inconsistency and semantic distortion.

- We perform comprehensive evaluations across multiple datasets and architectures, demonstrating the effectiveness and generalizability of SBMG and establishing a strong technical baseline for future multimodal motion planning research.

## 2 RELATED WORK

### 2.1 CONDITIONAL MOTION SYNTHESIS

In recent years, significant progress has been made in human motion generation techniques under conditional control Conditional motion synthesis typically leverages multimodal inputs and encompasses various control modalities, such as text (Guo et al., 2022; Kim et al., 2023; Lu et al., 2023; Petrovich et al., 2022; Zhou & Wang, 2023), audio (Wang et al., 2025; Kucherenko et al., 2019; Li et al., 2021), music (Li et al., 2024; Tseng et al., 2023), objects (Ghosh et al., 2023; Kulkarni et al., 2024; Li et al., 2023; Pi et al., 2023; Xu et al., 2023), and trajectories (Dai et al., 2024; Huang et al., 2024; Karunratanakul et al., 2024; Pinyoanuntapong et al., 2024a; Wan et al., 2024; Xie et al., 2023). These methods focus on creating realistic, context-specific motions by mapping inputs to motion parameters.

For trajectory control, methods like PriorMDM (Shafir et al., 2023) refine the MDM (Tevet et al., 2022b) model to achieve end-effector position control, while GMD (Karunratanakul et al., 2023) and Trace and Pace (Rempe et al., 2023) enable spatial control during diffusion by guiding the root joint positions. OmniControl (Xie et al., 2023) extends control to arbitrary joints, and MotionLCM (Dai et al., 2024) incorporates this control into latent spaces using ControlNet (Zhang et al., 2023a). Additionally, DNO (Karunratanakul et al., 2024) optimizes the diffusion noise process to generate motions aligning with differentiable objective functions.

### 2.2 TEXT-DRIVEN MOTION GENERATION

Text-driven motion generation in its early stages relied primarily on aligning latent distributions between motion and language, often implemented using loss functions like Kullback-Leibler (KL) divergence and contrastive loss. Representative works include Language2Pose (Ahuja & Morency, 2019), TEMOS (Petrovich et al., 2022), and MotionCLIP (Tevet et al., 2022a). However, due to inherent differencesbetween text and motion distributions, such latent space alignment methods are constrained in their ability to generate high-quality outputs.

Diffusion models have recently emerged as the dominant approach for text-to-motion generation, allowing researchers to denoise and synthesize complete motion sequences in motion spaces (Tevet et al., 2022a; Zhang et al., 2024), and quantized spaces (Lou et al., 2023). Additionally, token-based autoregressive models, such as GPT-inspired frameworks (Guo et al., 2022) and masked motion modeling (Guo et al., 2024; Hosseyni et al., 2025; Pinyoanuntapong et al., 2024c; Guo et al., 2025), have shown notable progress by learning to generate discrete motion token sequences with pretrained motion VQVAE (Williams et al., 2021; Esser et al., 2021; Williams et al., 2020).

Despite significant advances in conditional and text-driven motion generation, language-based control often remains coarse-grained, and multimodal models may produce inconsistent motions when input conflicts or is ambiguous. In this work, we investigate the misalignment between textual inputs and trajectory controls, particularly under conditions of dense or sparse semantics in the text, and we propose a novel modulation framework to enhance alignment in motion synthesis, yielding results that better satisfy user expectations.

## 3 METHOD

### 3.1 SIGNAL DENSITY MEASURING MODULE

#### 3.1.1 BASIC INFORMATION DENSITY

We approach from the perspective of temporal representations of signals, aiming to quantify the semantic information density of multimodal signals at each time step and provide effective guidance for motion generation through dynamic adjustment strategies. SDM extracts the information

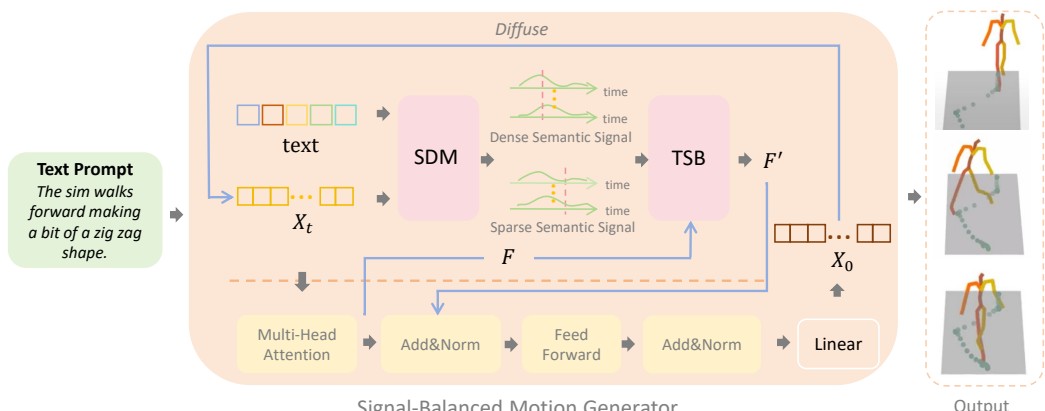

Figure 2: Overview of the SBMG Model. Input text description is first processed by our SDM module, generating dense and sparse semantic signals. These signals are then passed to the TSB module, which dynamically adjusts the intermediate motion feature $F$ to produce the modulated feature $F'$, resulting in the desired motion sequence.

density distribution across multiple time points by analyzing the alignment degree between different semantic signals and motion features, to provide precise adjustment signals for the subsequent generation process. Specifically, the input text description is first tokenized into a vocabulary sequence $\{v_1, v_2, \ldots, v_n\}$, and then, for each time step $t$, we compute the attention score $s$ between each token feature and the motion feature, which measures the degree of matching between the word features and motion features at each time step, as shown below:

$$s = \text{softmax}\left(\frac{EX^T}{\sqrt{d_k}}\right) X \tag{1}$$

where $E$ represents the encoded text embedding, $X$ represents the motion features, and $d_k$ represents the feature dimension of $X$.

To detect dense and sparse semantic scenarios, we analyze the information density distribution of each signal at different time points. From $s$, we extract the minimum and maximum values at each time step and apply mean filtering to them respectively, to retain important signal information.

If the attention distribution of vocabulary features over motion features is highly concentrated at a given time step, meaning that at this time step, the minimum value of attention across all vocabulary-to-motion feature pairs exceeds a threshold, then the time step is marked with a dense semantic signal. We define the dense semantic signal $\text{sig}_{\text{base-dense}}$ as the set of attention values that satisfy the following conditions:

$$\text{sig}_{\text{base-dense}}(t) = \frac{\min(s(t))}{\left(\min(s(t)) \leq \frac{1}{T} \sum_{t=1}^{T} \min(s(t))\right) + 1} \tag{2}$$

Where $\min(s(t))$ denotes the minimum attention value at time step $t$, and $\frac{1}{T} \sum_{t=1}^{T} \min(s(t))$ represents the threshold, which also denotes the average of the minimum attention values across all time steps.

When the attention distribution of vocabulary features over motion features is sparse at a given time step, meaning that at this time step, the maximum attention value across all vocabulary-to-motion feature pairs is smaller than the threshold, the time step is marked with a sparse semantic signal. We define the sparse semantic signal $\text{sig}_{\text{base-sparse}}$ as the set of attention values that satisfy the following conditions:

$$\text{sig}_{\text{base-sparse}}(t) = \frac{\max(s(t))}{\left(\max(s(t)) \geq \frac{1}{T} \sum_{t=1}^{T} \max(s(t))\right) + 1} \tag{3}$$

Where $\max(s(t))$ denotes the maximum attention value at time step $t$, and $\frac{1}{T} \sum_{t=1}^{T} \max(s(t))$

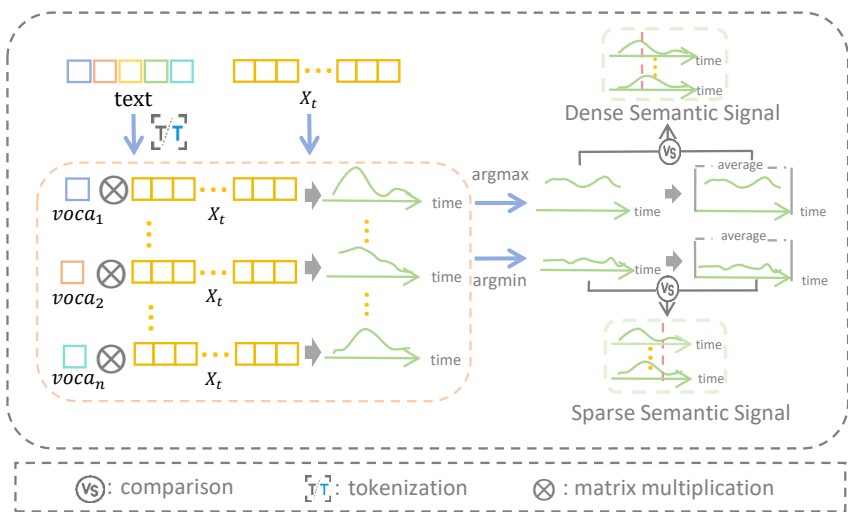

Figure 3: SDM Module Details. This module accepts text and motion inputs to generate the Dense Semantic Signal and Sparse Semantic Signal, which are used for dynamic adjustment in subsequent motion generation.

represents the threshold, which also denotes the average of the maximum attention values across all time steps.

### 3.1.2 COMPOSITE INFORMATION DENSITY

However, the basic information density, based solely on the matching strength at the current time step, is insufficient to reflect the dynamic changes of semantics over time fully. To more accurately capture the temporal characteristics of semantic information during motion generation, we combine the temporal variations of attention with the basic information density, forming the criteria for composite information density.

In this process, we extract the minimum value $A_{\text{low}}$ and maximum value $A_{\text{high}}$ of attention at each time step, forming two attention sequences. The minimum attention represents the "weakest" alignment strength among all vocabulary at this time step. If it is still above the threshold, it indicates that the overall semantic signal is relatively dense. The maximum attention represents the "strongest" alignment strength among all vocabulary at this time step. If it is still below the threshold, it suggests that the overall semantic signal is relatively sparse. Then, we compute the changes between adjacent time steps to model the temporal variation of information density in the sequence.

$$
\begin{aligned}
\Delta A_{\text{high}} &= A_{\text{high}} - \text{roll}(A_{\text{high}}, 1), \\
\Delta A_{\text{low}} &= A_{\text{low}} - \text{roll}(A_{\text{low}}, 1).
\end{aligned}
\tag{4}
$$

Here, the function $\text{roll}(\cdot, 1)$ represents shifting the sequence by one time step along the time dimension to simulate a sliding effect, thereby calculating the change between adjacent time steps and effectively capturing the variation in information density across the sequence. To identify time steps with significant changes, we take the absolute value of the difference, set its mean as the threshold, and filter out the time periods with significant semantic intensity changes, thus obtaining the dense and sparse semantic signals:

$$
\begin{aligned}
\text{sig}_{\text{diff-sparse}}(t) &= \begin{cases} A_{\text{high}}, & \text{if } |\Delta A_{\text{high}}| < \mu |\Delta A_{\text{high}}| \\ 0, & \text{otherwise} \end{cases} \\
\text{sig}_{\text{diff-dense}}(t) &= \begin{cases} A_{\text{low}}, & \text{if } |\Delta A_{\text{low}}| > \mu |\Delta A_{\text{low}}| \\ 0, & \text{otherwise} \end{cases}
\end{aligned}
\tag{5}
$$

This module not only captures the semantic density at the current time step but also sensitively reflects the dynamic changes in semantic information, providing a reference with enhanced temporal

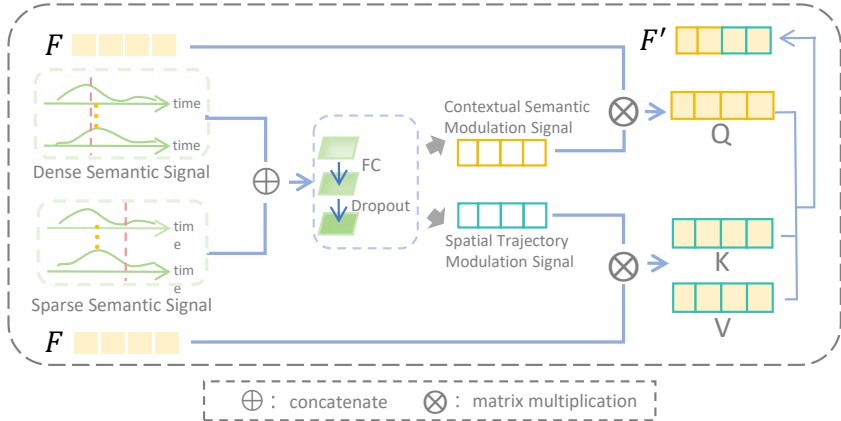

Figure 4: TSB Module Details. This module accepts Dense Semantic Signal, Sparse Semantic Signal, and intermediate motion feature $F$ as inputs, generating the dynamically adjusted $F'$ to produce motions aligned with user intent.

recognition capability for the signal balancing strategy during the generation process. To fully integrate the basic information density and the variation information across the time dimension, the composite information density is adaptively weighted and fused by learnable parameters $\alpha$ and $\beta$:

$$\begin{aligned}
\text{sig}_{\text{dense}} &= \alpha \cdot \text{sig}_{\text{base-dense}} + \beta \cdot \text{sig}_{\text{diff-dense}}, \\
\text{sig}_{\text{sparse}} &= \alpha \cdot \text{sig}_{\text{base-sparse}} + \beta \cdot \text{sig}_{\text{diff-sparse}}.
\end{aligned} \tag{6}$$

Here, $\text{sig}_{\text{dense}}$ denotes the dense semantic signal, and $\text{sig}_{\text{sparse}}$ denotes the sparse semantic signal. With initial values set to 0.5, $\alpha$ and $\beta$ are optimized together with the model parameters, enabling the system to balance temporal variations in semantic information with the overall intensity distribution, thus providing more refined modulation signals for the generation process.

In summary, through dense and sparse semantic signals, SDM effectively provides adjustment signals for motion generation. These signals are then processed in the TSB to generate more accurate dynamic modulation signals, thus improving the coordinated control between text and trajectory.

## 3.2 TEXT-TRAJECTORY SIGNAL BALANCING STRUCTURE

Although the signal density measurement module effectively identifies densely and sparsely populated semantic signals, dynamically adjusting the weighting of text and trajectory signals during generation to achieve precise semantic and trajectory control remains challenging. The primary task of TSB is to generate balanced embedding representations at different timesteps based on the signals from SDM, thus enabling the dynamic modulation of text and trajectory signals. By developing a Contextual Semantic Modulation Signal and a Spatial Trajectory Modulation Signal, the model coordinates the guiding influence of each signal on the generated motion, enhancing sequence coherence and accuracy.

In the text-trajectory signal balancing structure, $\text{sig}_{\text{dense}}$, $\text{sig}_{\text{sparse}}$, and $x_{\text{seq}}$ are first abstracted into multi-scale features and fused. These fused features are then concatenated into a combined feature vector. The combined vector is transformed through a linear layer with a Sigmoid activation function to produce a modulation factor $g$.

$$g = \sigma(W \cdot \text{combined} + b) \tag{7}$$

where $\sigma$ denotes the Sigmoid activation function, and $W$ and $b$ are the weight matrix and bias term of the linear transformation, respectively.

The modulation factor controls the generation of the Contextual Semantic Modulation Signal and Spatial Trajectory Modulation Signal. These signals are subsequently fused with the motion feature $F$ within a mask-defined area. The fused features are then processed through a cross-attention mechanism to further adjust the influence of text and trajectory signals, resulting in dynamically

optimized motion features $F'$. This balancing process significantly enhances the model's capability to handle densely and sparsely populated semantics, ensuring the generated motion sequence aligns more closely with user expectations.

## 4 EXPERIMENTS

### 4.1 DATASETS.

We conduct experiments on two widely used public text-motion datasets: the HumanML3D dataset (Guo et al., 2022) and the KIT-ML dataset (Plappert et al., 2016). The HumanML3D dataset comprises 14,616 unique motion capture sequences sourced from the AMASS (Mahmood et al., 2019) and HumanAct12 (Guo et al., 2020) datasets, along with 44,970 corresponding text descriptions. It encompasses a wide range of human activities, including locomotion, sports, and acrobatics. The KIT-ML dataset, though smaller in scale, contains 3,911 motion sequences paired with 6,278 text descriptions. Both datasets are preprocessed and split into training, validation, and test sets according to the procedure described in Guo et al. (2022).

### 4.2 EVALUATION METRICS.

To comprehensively evaluate the quality and control performance of motion sequences generated by the three backbone networks, we employed several evaluation metrics. For OmniControl, Fréchet Inception Distance (FID) was used to assess the naturalness and fidelity of generated motions. R-Precision was utilized to evaluate the relevance between motions and their corresponding text prompts, while the Diversity metric quantified the variation in generated motions. Additionally, 3D control error metrics such as trajectory error, position error, and average error were used to measure the control precision of keyframe joint positions.

For MoMask and SALAD, we adopted similar metrics, including FID, R-Precision, and Diversity. To further assess semantic alignment, we introduced Multimodal Distance (MM-Dist) to evaluate how well the generated motions semantically match the input text, and employed Multimodality (MModality) to measure the diversity of motions generated from the same prompt. We emphasize the importance of balancing multimodality with quality metrics, as over-optimizing for diversity may compromise the consistency and relevance of generated results.

### 4.3 COMPARISON WITH OTHER METHODS

To validate the effectiveness of the method proposed in this paper, we conducted experiments on three backbone networks, OmniControl (Xie et al., 2023), MoMask (Guo et al., 2024), and SALAD (Hong et al., 2025), respectively, and compared them with current mainstream methods. Due to the unique evaluation metrics of OmniControl, we summarize its results separately in Section A.3, where the data for OmniControl is based on our reproduced model. When comparing with MDM (Tevet et al., 2022b), PriorMDM (Shafir et al., 2023), and GMD (Karunratanakul et al., 2023), we focus solely on pelvis control to ensure fairness in the comparisons. To further validate the multi-joint control performance of our method, we also conducted comparisons with OmniControl under additional combinations of control joints.

Considering that MoMask and SALAD follow a shared evaluation framework, we perform comparisons on the HumanML3D and KIT-ML datasets against several state-of-the-art methods, including T2M (Guo et al., 2022), MDM (Tevet et al., 2022b), BAMM (Pinyoanuntapong et al., 2024b), ReMoDiffuse (Zhang et al., 2023b), MoMask (Guo et al., 2024), SALAD (Hong et al., 2025), Mo-GenTS (Yuan et al., 2024), and Motion Anything (Zhang et al., 2025). Each experiment is repeated 20 times, and the reported results include the mean and the 95% confidence interval. The experimental results are summarized in Table 1. Qualitative evaluation results are provided in Section A.5.

Table 1: Quantitative comparison of MoMask and SALAD with other methods on HumanML3D and KIT-ML. The best and runner-up values are bold and underlined. The right arrow $\rightarrow$ indicates that closer values to ground truth are better.

| Datasets | Methods | R Precision↑ | | | FID↓ | MM-Dist↓ | Diversity→ | MultiModality↑ |
|---|---|---|---|---|---|---|---|---|
| | | Top 1 | Top 2 | Top 3 | | | | |
| Human ML3D | Ground Truth | $0.511^{\pm.003}$ | $0.703^{\pm.003}$ | $0.797^{\pm.002}$ | $0.002^{\pm.000}$ | $2.974^{\pm.008}$ | $9.503^{\pm.065}$ | - |
| | T2M | $0.455^{\pm.003}$ | $0.636^{\pm.003}$ | $0.736^{\pm.002}$ | $1.087^{\pm.021}$ | $3.347^{\pm.008}$ | $9.175^{\pm.083}$ | $2.219^{\pm.074}$ |
| | MDM | $0.320^{\pm.005}$ | $0.498^{\pm.004}$ | $0.611^{\pm.007}$ | $0.544^{\pm.044}$ | $5.566^{\pm.027}$ | $9.559^{\pm.086}$ | $\mathbf{2.799}^{\pm.072}$ |
| | BAMM | $0.525^{\pm.002}$ | $0.720^{\pm.003}$ | $0.814^{\pm.003}$ | $0.055^{\pm.002}$ | $2.919^{\pm.008}$ | $9.717^{\pm.089}$ | $1.687^{\pm.051}$ |
| | ReMoDiffuse | $0.510^{\pm.005}$ | $0.698^{\pm.006}$ | $0.795^{\pm.004}$ | $0.103^{\pm.004}$ | $2.974^{\pm.016}$ | $9.018^{\pm.075}$ | $1.795^{\pm.043}$ |
| | MoMask | $0.521^{\pm.002}$ | $0.713^{\pm.002}$ | $0.807^{\pm.002}$ | $0.045^{\pm.002}$ | $2.958^{\pm.008}$ | - | $1.241^{\pm.040}$ |
| | SALAD | $\underline{0.581}^{\pm.003}$ | $\underline{0.769}^{\pm.003}$ | $\underline{0.857}^{\pm.002}$ | $0.076^{\pm.002}$ | $\underline{2.649}^{\pm.009}$ | $9.696^{\pm.096}$ | $1.751^{\pm.062}$ |
| | MoGenTS | $0.529^{\pm.003}$ | $0.719^{\pm.002}$ | $0.812^{\pm.002}$ | $0.033^{\pm.001}$ | $2.867^{\pm.006}$ | $\underline{9.570}^{\pm.077}$ | - |
| | Motion Anything | $0.546^{\pm.003}$ | $0.735^{\pm.002}$ | $0.829^{\pm.002}$ | $\mathbf{0.028}^{\pm.005}$ | $2.859^{\pm.010}$ | $\mathbf{9.521}^{\pm.083}$ | $\underline{2.705}^{\pm.068}$ |
| | MoMask (Ours) | $0.532^{\pm.003}$ | $0.719^{\pm.002}$ | $0.812^{\pm.002}$ | $\underline{0.030}^{\pm.001}$ | $2.938^{\pm.004}$ | - | $1.283^{\pm.048}$ |
| | SALAD (Ours) | $\mathbf{0.583}^{\pm.003}$ | $\mathbf{0.774}^{\pm.003}$ | $\mathbf{0.859}^{\pm.002}$ | $0.074^{\pm.002}$ | $\mathbf{2.637}^{\pm.007}$ | $9.660^{\pm.086}$ | $1.756^{\pm.074}$ |
| KIT-ML | Ground Truth | $0.424^{\pm.005}$ | $0.649^{\pm.006}$ | $0.779^{\pm.006}$ | $0.031^{\pm.004}$ | $2.788^{\pm.012}$ | $11.08^{\pm.097}$ | - |
| | T2M | $0.361^{\pm.005}$ | $0.559^{\pm.007}$ | $0.681^{\pm.007}$ | $3.022^{\pm.107}$ | $3.488^{\pm.028}$ | $10.72^{\pm.145}$ | $\mathbf{2.052}^{\pm.107}$ |
| | MDM | $0.164^{\pm.004}$ | $0.291^{\pm.004}$ | $0.396^{\pm.004}$ | $0.497^{\pm.021}$ | $9.191^{\pm.022}$ | $10.847^{\pm.109}$ | $\underline{1.907}^{\pm.214}$ |
| | BAMM | $0.438^{\pm.009}$ | $0.661^{\pm.009}$ | $0.788^{\pm.005}$ | $0.183^{\pm.013}$ | $2.723^{\pm.026}$ | $11.01^{\pm.094}$ | $1.609^{\pm.065}$ |
| | ReMoDiffuse | $0.427^{\pm.014}$ | $0.641^{\pm.004}$ | $0.765^{\pm.055}$ | $0.155^{\pm.006}$ | $2.814^{\pm.012}$ | $10.80^{\pm.105}$ | $1.239^{\pm.028}$ |
| | MoMask | $0.433^{\pm.007}$ | $0.656^{\pm.005}$ | $0.781^{\pm.005}$ | $0.204^{\pm.011}$ | $2.779^{\pm.022}$ | - | $1.131^{\pm.043}$ |
| | SALAD | $\underline{0.477}^{\pm.006}$ | $\underline{0.711}^{\pm.005}$ | $\mathbf{0.828}^{\pm.005}$ | $0.296^{\pm.012}$ | $\underline{2.585}^{\pm.016}$ | $\mathbf{11.097}^{\pm.095}$ | $1.004^{\pm.040}$ |
| | MoGenTS | $0.445^{\pm.006}$ | $0.671^{\pm.006}$ | $0.797^{\pm.005}$ | $0.143^{\pm.004}$ | $2.711^{\pm.024}$ | $10.92^{\pm.090}$ | - |
| | Motion Anything | $0.449^{\pm.007}$ | $0.678^{\pm.004}$ | $\underline{0.802}^{\pm.006}$ | $\mathbf{0.131}^{\pm.003}$ | $2.705^{\pm.024}$ | $10.94^{\pm.098}$ | $1.374^{\pm.069}$ |
| | MoMask (Ours) | $0.441^{\pm.005}$ | $0.665^{\pm.006}$ | $0.788^{\pm.007}$ | $\underline{0.141}^{\pm.020}$ | $2.757^{\pm.014}$ | - | $1.149^{\pm.042}$ |
| | SALAD (Ours) | $\mathbf{0.482}^{\pm.007}$ | $\mathbf{0.713}^{\pm.006}$ | $\mathbf{0.828}^{\pm.005}$ | $0.249^{\pm.015}$ | $\mathbf{2.558}^{\pm.017}$ | $11.174^{\pm.011}$ | $0.963^{\pm.032}$ |

## 4.4 ABLATION STUDIES

### 4.4.1 IMPACT OF SEMANTIC DENSITY PERCEPTION ON GENERATION PERFORMANCE.

In this experiment, we explore the impact of different semantic density judgment strategies on generation performance. Our approach dynamically captures the trends of signal changes and combines temporal information to accurately identify high and low-density regions, thereby optimizing signal weights. Unlike traditional density evaluation based on statistical features, our method intelligently responds to signal fluctuations, significantly improving generation performance. The results of this experiment are presented in Table 2.

### 4.4.2 IMPACT OF MODULATION SIGNAL FUSION METHODS ON GENERATION PERFORMANCE.

This experiment investigates the application of a multi-head attention mechanism and multi-scale pooling in signal fusion. The multi-head attention mechanism optimizes computational resources by processing signals in parallel, especially during time steps with higher information density, while multi-scale pooling further refines signal processing by pooling intermediate results. Experimental results show that although these two methods independently improve generation performance, their precision and quality still fall short compared to our approach. Our method significantly enhances generation quality through more precise signal modulation and dynamic control. The results of this experiment are presented in Table 3.

### 4.4.3 IMPACT OF INFORMATION DENSITY THRESHOLD STRATEGIES ON GENERATION PERFORMANCE

This experiment evaluates the effect of different threshold-setting strategies in the SDM for detecting dense and sparse semantic signals. We compare fixed threshold, global adaptive threshold, and our proposed time-series adaptive threshold. Results show that the fixed threshold struggles to generalize across motions of varying complexity, while our time-series adaptive threshold achieves a better balance between global and local variations, leading to more accurate identification of dense/sparse signals and significantly improving motion continuity and control precision in the final generation results. The results of this experiment are presented in Table 4.

Table 2: Ablation study of the semantic density perception strategy on generation performance.

| Methods | R Precision↑ | | | FID↓ | MM-Dist↓ | MultiModality↑ |
|---|---|---|---|---|---|---|
| | Top 1 | Top 2 | Top 3 | | | |
| Standard Deviation | $0.519^{\pm.003}$ | $0.709^{\pm.002}$ | $0.804^{\pm.002}$ | $0.044^{\pm.002}$ | $2.979^{\pm.009}$ | $1.366^{\pm.041}$ |
| Entropy | $0.483^{\pm.002}$ | $0.674^{\pm.003}$ | $0.775^{\pm.002}$ | $0.136^{\pm.007}$ | $3.181^{\pm.009}$ | $\mathbf{1.651}^{\pm.050}$ |
| Median | $0.516^{\pm.003}$ | $0.710^{\pm.002}$ | $0.806^{\pm.002}$ | $0.047^{\pm.002}$ | $2.968^{\pm.008}$ | $1.413^{\pm.041}$ |
| Quantile | $0.519^{\pm.003}$ | $0.712^{\pm.003}$ | $0.808^{\pm.002}$ | $0.035^{\pm.002}$ | $2.945^{\pm.007}$ | $1.320^{\pm.039}$ |
| **Ours** | $\mathbf{0.532}^{\pm.003}$ | $\mathbf{0.719}^{\pm.002}$ | $\mathbf{0.812}^{\pm.002}$ | $\mathbf{0.030}^{\pm.001}$ | $\mathbf{2.938}^{\pm.004}$ | $1.283^{\pm.048}$ |

Table 3: Ablation study of the modulation signal fusion methods on generation performance.

| Methods | R Precision↑ | | | FID↓ | MM-Dist↓ | MultiModality↑ |
|---|---|---|---|---|---|---|
| | Top 1 | Top 2 | Top 3 | | | |
| Multi-head Attention | $0.521^{\pm.003}$ | $0.712^{\pm.002}$ | $0.804^{\pm.002}$ | $0.042^{\pm.002}$ | $2.954^{\pm.010}$ | $1.272^{\pm.061}$ |
| Multi-scale Pooling | $0.530^{\pm.003}$ | $\mathbf{0.721}^{\pm.002}$ | $0.811^{\pm.002}$ | $0.042^{\pm.002}$ | $\mathbf{2.914}^{\pm.007}$ | $\mathbf{1.329}^{\pm.046}$ |
| **Ours** | $\mathbf{0.532}^{\pm.003}$ | $0.719^{\pm.002}$ | $\mathbf{0.812}^{\pm.002}$ | $\mathbf{0.030}^{\pm.001}$ | $2.938^{\pm.004}$ | $1.283^{\pm.048}$ |

Table 4: Ablation study of the information density threshold strategies on generation performance.

| Methods | R Precision↑ | | | FID↓ | MM-Dist↓ | MultiModality↑ |
|---|---|---|---|---|---|---|
| | Top 1 | Top 2 | Top 3 | | | |
| Fixed Threshold | $0.520^{\pm.003}$ | $0.710^{\pm.002}$ | $0.801^{\pm.002}$ | $0.044^{\pm.002}$ | $2.965^{\pm.009}$ | $1.255^{\pm.055}$ |
| Global Adaptive Threshold | $0.526^{\pm.003}$ | $0.716^{\pm.002}$ | $0.808^{\pm.002}$ | $0.038^{\pm.001}$ | $2.945^{\pm.006}$ | $1.271^{\pm.050}$ |
| **Ours** | $\mathbf{0.532}^{\pm.003}$ | $\mathbf{0.719}^{\pm.002}$ | $\mathbf{0.812}^{\pm.002}$ | $\mathbf{0.030}^{\pm.001}$ | $\mathbf{2.938}^{\pm.004}$ | $\mathbf{1.283}^{\pm.048}$ |

Table 5: Ablation study of the weighting strategies on generation performance.

| Methods | R Precision↑ | | | FID↓ | MM-Dist↓ | MultiModality↑ |
|---|---|---|---|---|---|---|
| | Top 1 | Top 2 | Top 3 | | | |
| Fixed Weighting | $0.521^{\pm.003}$ | $0.711^{\pm.002}$ | $0.803^{\pm.002}$ | $0.043^{\pm.002}$ | $2.960^{\pm.008}$ | $1.262^{\pm.053}$ |
| Simple Proportional Weighting | $0.527^{\pm.003}$ | $0.715^{\pm.002}$ | $0.808^{\pm.002}$ | $0.037^{\pm.001}$ | $2.944^{\pm.007}$ | $1.275^{\pm.047}$ |
| **Ours** | $\mathbf{0.532}^{\pm.003}$ | $\mathbf{0.719}^{\pm.002}$ | $\mathbf{0.812}^{\pm.002}$ | $\mathbf{0.030}^{\pm.001}$ | $\mathbf{2.938}^{\pm.004}$ | $\mathbf{1.283}^{\pm.048}$ |

### 4.4.4 IMPACT OF WEIGHTING STRATEGIES ON GENERATION PERFORMANCE

This experiment investigates the impact of different weighting strategies when integrating basic information density and temporal variation information. We compare a learnable dynamic weighting method with two baseline approaches: fixed weighting and simple proportional weighting. The experimental results show that, compared to the baselines, the learnable dynamic weighting strategy achieves superior performance in terms of motion continuity, semantic consistency, and control precision. The results of this experiment are presented in Table 5. Additional ablation studies complementing are presented in Section A.4

## 5 CONCLUSION

In conclusion, the Signals-Balanced Motion Generator (SBMG) introduces an effective solution for multimodal motion generation by dynamically balancing text and trajectory signals. By incorporating a Signal Density Measuring Module and a Text-Trajectory Signal Balancing Block, SBMG addresses the challenges of aligning generated motions with user expectations, mitigating action inconsistencies and semantic misalignments. Our extensive experiments demonstrate SBMG's superior performance in both qualitative and quantitative assessments, highlighting its potential in advancing the field of motion generation.

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

## A APPENDIX

### A.1 THE USE OF LLMS

In this work, LLMs were used solely as general-purpose tools to aid and polish the writing of the manuscript. The ideas, scientific content, experimental design, and other related aspects were developed entirely by the authors.

### A.2 NETWORK OVERVIEW

In the task of multimodal motion generation, text descriptions and trajectory signals often exhibit uneven and dynamically changing information density in the time dimension. When the model fails to identify these fluctuations in semantic density accurately, it is prone to erroneously assigning excessively high or low weights to a particular modal signal at moments of overly dense or sparse semantics, leading to semantic disorder and poor continuity in the generated motions. The root cause of this disharmony lies in the fact that existing methods generally lack accurate quantification

and dynamic regulation mechanisms for the semantic information density of different modal signals in the temporal sequence, thus failing to adapt to the fluctuations and imbalances in the information content of text and trajectory signals across different periods.

To delve deeper into this issue, we approach it from the perspective of the semantic density of input signals, arguing that the importance of each modal signal in different time steps during the motion generation process should be dynamically adjusted based on the actual effective information it carries. Specifically, the information density of text and trajectory signals on the time axis exhibits highs and lows; if the model blindly adopts static or simple weighted fusion, it is difficult to achieve temporal alignment and reasonable balance of semantics, resulting in motion planning that may overly rely on text while ignoring trajectory cues during semantically dense periods, and vice versa.

Therefore, this study proposes a dynamic signal-balanced generation framework driven by information density, aiming to precisely capture the changes in semantic intensity of multimodal signals in the temporal sequence and adjust the guiding weights of each modality in the motion generation process accordingly. The framework consists of two key components: first, by analyzing the matching degree between text and motion features, quantify the information density of multimodal signals at different time steps, revealing semantically dense and sparse intervals; second, based on this information density distribution, dynamically generate control signals that reflect the balance state of semantics and trajectories, achieving cross-modal temporal alignment and adaptive weight adjustment, thereby mitigating the negative impacts of signal conflicts on motion generation.

### A.3 DETAILED EXPERIMENTAL RESULTS OF OMNICONTROL

Due to OmniControl (Xie et al., 2023) employing different evaluation metrics and settings compared to other methods, its experimental results are not included in the main text but are compiled separately in this section. Tables 6 and 7 show the experimental results of OmniControl.

Table 6: Quantitative results of the first backbone network (OmniControl) on the HumanML3D test set. Best results are in bold.

| Method | Control Joint | FID↓ | R-precision↑ (Top-3) | Diversity→ | Traj. Err.↓ (50 cm) | Loc. Err.↓ (50 cm) | Avg. Err. (cm)↓ |
|---|---|---|---|---|---|---|---|
| Real | - | 0.002 | 0.797 | 9.503 | 0.000 | 0.000 | 0.000 |
| MDM | | 0.698 | 0.602 | 9.197 | 0.402 | 0.308 | 0.596 |
| PriorMDM | Pelvis | 0.475 | 0.583 | 9.156 | 0.346 | 0.213 | 0.442 |
| GMD | | 0.576 | 0.665 | 9.206 | 0.931 | 0.032 | 0.144 |
| OmniControl | | 0.387 | **0.712** | 9.705 | 0.136 | 0.024 | 0.073 |
| Ours (on pelvis) | | **0.104** | 0.705 | **9.352** | **0.033** | **0.003** | **0.031** |
| OmniControl | Pelvis | 0.323 | 0.691 | 9.854 | 0.078 | 0.013 | 0.064 |
| Ours (on all) | | **0.155** | **0.694** | **9.447** | **0.041** | **0.005** | **0.048** |
| OmniControl | Head | 0.316 | 0.687 | 9.921 | 0.093 | 0.017 | 0.076 |
| Ours (on all) | | **0.158** | **0.703** | **9.435** | **0.057** | **0.008** | **0.064** |
| OmniControl | Left Hand | 0.264 | 0.690 | 9.661 | 0.200 | 0.028 | 0.119 |
| Ours (on all) | | **0.122** | **0.702** | **9.331** | **0.139** | **0.015** | **0.096** |
| OmniControl | Right Hand | 0.264 | 0.690 | 9.661 | 0.200 | 0.028 | 0.119 |
| Ours (on all) | | **0.127** | **0.696** | **9.359** | **0.129** | **0.014** | **0.093** |
| OmniControl | Left Foot | 0.292 | 0.689 | 9.855 | 0.123 | 0.017 | 0.062 |
| Ours (on all) | | **0.165** | **0.690** | **9.593** | **0.101** | **0.010** | **0.051** |
| OmniControl | Right Foot | 0.307 | 0.693 | 9.901 | 0.143 | 0.019 | 0.065 |
| Ours (on all) | | **0.135** | **0.713** | **9.511** | **0.109** | **0.013** | **0.055** |
| OmniControl | Average | 0.297 | 0.691 | 9.828 | 0.139 | 0.019 | 0.083 |
| Ours (on all) | | **0.144** | **0.700** | **9.446** | **0.096** | **0.011** | **0.068** |

### A.4 ADDITIONAL ABLATION STUDIES

#### A.4.1 IMPACT OF SEMANTIC DENSITY VARIATION ON GENERATION PERFORMANCE.

We further investigate the impact of semantic density variation on generation performance. By combining the judgment of base semantic density with time step variations, we can precisely identify key change moments and dynamically adjust signal weights. This dual semantic density judgment strat-

Table 7: Quantitative results of the first backbone network (OmniControl) on the KIT test set. Best results are in bold.

| Method | Control Joint | FID↓ | R-precision↑ (Top-3) | Diversity→ | Traj. Err.↓ (50 cm) | Loc. Err.↓ (50 cm) | Avg. Err. (cm)↓ |
|---|---|---|---|---|---|---|---|
| Real | - | 0.031 | 0.779 | 11.08 | 0.000 | 0.000 | 0.000 |
| PriorMDM | | 0.851 | 0.397 | 10.518 | 0.3310 | 0.1400 | 0.2305 |
| GMD | Pelvis | 1.565 | 0.382 | 9.664 | 0.5443 | 0.3003 | 0.4070 |
| OmniControl | | 0.702 | 0.397 | 10.927 | 0.1105 | 0.0337 | **0.0759** |
| Ours | | 0.994 | 0.399 | **11.123** | 0.1619 | 0.0358 | 0.1100 |
| OmniControl | Average | 0.788 | 0.379 | 10.841 | 0.1433 | 0.0368 | 0.0854 |
| Ours | | **0.665** | **0.411** | 11.258 | **0.1295** | **0.0261** | 0.0948 |

Table 8: Ablation study of the impact of semantic density variation on generation performance.

| Methods | R Precision↑ | | | FID↓ | MM-Dist↓ | MultiModality↑ |
|---|---|---|---|---|---|---|
| | Top 1 | Top 2 | Top 3 | | | |
| Temporal Difference | $0.525^{\pm.003}$ | $0.717^{\pm.002}$ | $0.810^{\pm.003}$ | $0.032^{\pm.002}$ | $\mathbf{2.931}^{\pm.006}$ | $1.165^{\pm.051}$ |
| **Ours** | $\mathbf{0.532}^{\pm.003}$ | $\mathbf{0.719}^{\pm.002}$ | $\mathbf{0.812}^{\pm.002}$ | $\mathbf{0.030}^{\pm.001}$ | $2.938^{\pm.004}$ | $\mathbf{1.283}^{\pm.048}$ |

Table 9: Ablation study of gating mechanisms on signal fusion effectiveness.

| Methods | R Precision↑ | | | FID↓ | MM-Dist↓ | MultiModality↑ |
|---|---|---|---|---|---|---|
| | Top 1 | Top 2 | Top 3 | | | |
| Forgetting Gate | $0.526^{\pm.002}$ | $\mathbf{0.719}^{\pm.002}$ | $\mathbf{0.812}^{\pm.002}$ | $0.037^{\pm.002}$ | $\mathbf{2.918}^{\pm.007}$ | $1.312^{\pm.055}$ |
| Weighted Gate | $0.518^{\pm.002}$ | $0.712^{\pm.002}$ | $0.806^{\pm.002}$ | $0.041^{\pm.002}$ | $2.962^{\pm.007}$ | $\mathbf{1.386}^{\pm.063}$ |
| Nonlinear Gate | $0.518^{\pm.003}$ | $0.713^{\pm.003}$ | $0.808^{\pm.002}$ | $0.047^{\pm.002}$ | $2.950^{\pm.009}$ | $1.354^{\pm.061}$ |
| Low-rank Gate | $0.523^{\pm.003}$ | $0.716^{\pm.003}$ | $0.811^{\pm.002}$ | $0.043^{\pm.002}$ | $2.938^{\pm.007}$ | $1.354^{\pm.061}$ |
| **Ours** | $\mathbf{0.532}^{\pm.003}$ | $\mathbf{0.719}^{\pm.002}$ | $\mathbf{0.812}^{\pm.002}$ | $\mathbf{0.030}^{\pm.001}$ | $2.938^{\pm.004}$ | $1.283^{\pm.048}$ |

Table 10: Ablation study of the dynamic sliding window on generation performance.

| Methods | R Precision↑ | | | FID↓ | MM-Dist↓ | MultiModality↑ |
|---|---|---|---|---|---|---|
| | Top 1 | Top 2 | Top 3 | | | |
| Dynamic Sliding Window | $0.520^{\pm.002}$ | $0.710^{\pm.002}$ | $0.805^{\pm.002}$ | $0.039^{\pm.002}$ | $2.967^{\pm.007}$ | $\mathbf{1.353}^{\pm.038}$ |
| **Ours** | $\mathbf{0.532}^{\pm.003}$ | $\mathbf{0.719}^{\pm.002}$ | $\mathbf{0.812}^{\pm.002}$ | $\mathbf{0.030}^{\pm.001}$ | $\mathbf{2.938}^{\pm.004}$ | $1.283^{\pm.048}$ |

egy effectively enhances generation accuracy and diversity, especially in complex temporal tasks, outperforming strategies that solely rely on time step variations. The results of this experiment are presented in Table 8.

### A.4.2 IMPACT OF GATING MECHANISMS ON SIGNAL FUSION EFFECTIVENESS.

We compare the impact of different gating mechanisms on signal fusion. While these mechanisms optimize signal weight adjustment to some extent, our research shows that with more refined gating adjustments, our method exhibits significant advantages in temporal control of dense and sparse signals, enhancing generation accuracy, quality, and robustness. The results of this experiment are presented in Table 9.

### A.4.3 IMPACT OF DYNAMIC SLIDING WINDOW ON GENERATION PERFORMANCE.

This experiment employs the dynamic sliding window method for signal density recognition and adjustment. By using sliding windows, we can capture local signal fluctuations at each time step and dynamically adjust signal density. Although this method improves generation control to some extent, its performance still has room for improvement compared to our method. Our approach significantly enhances generation quality through more precise signal modulation. The results of this experiment are presented in Table 10.

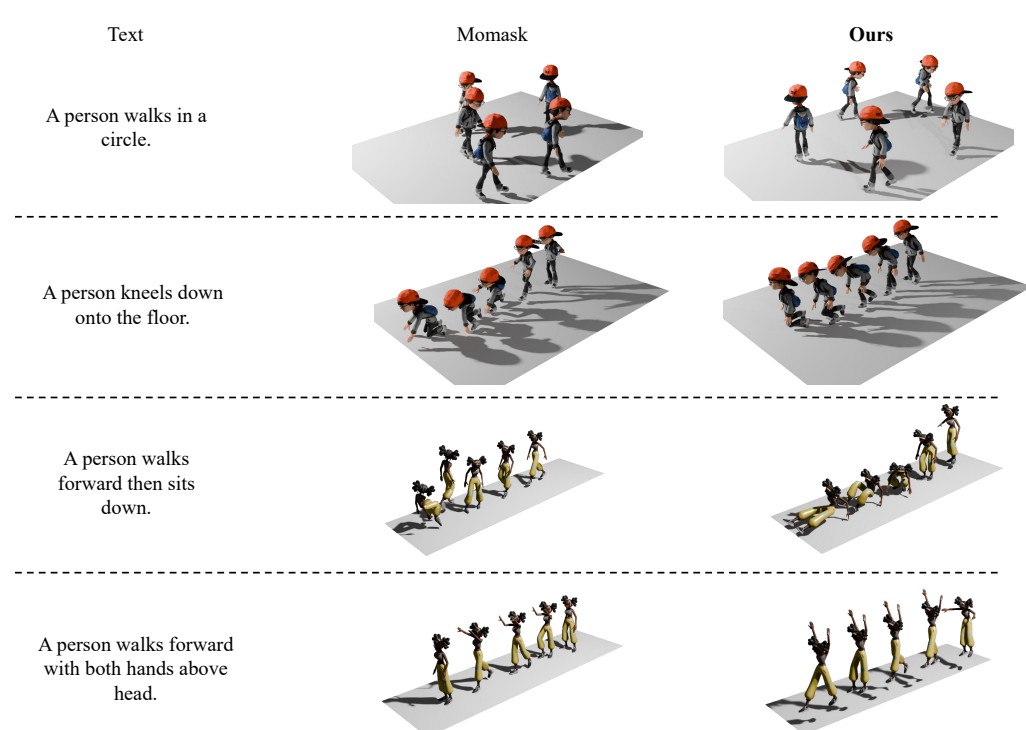

Figure 5: Qualitative evaluation of text-to-motion generation. We conducted a qualitative comparison between the motion visualizations generated by our method and those generated by Mo-Mask (Guo et al., 2024).

## A.5 QUALITATIVE EVALUATION

To qualitatively evaluate the performance of our text-to-motion generation method, We compare the visualizations produced by our approach with those generated by a representative existing method in text-to-motion generation, including MoMask (Guo et al., 2024). The text prompts are customized based on the HumanML3D test set. As shown in Figure 5, our method demonstrates significant advantages over the existing methods in terms of motion quality, diversity, and alignment between text and motion.

