# OpenReview forum: "Balance Your Move: Investigating Text to Motion from a Signal Density Measuring View"
_ICLR.cc/2026/Conference — ICLR 2026 Conference Withdrawn Submission_

### Official Review · Reviewer_oXUy · 2025-10-28

**Soundness:** 2
**Presentation:** 1
**Contribution:** 3
**Rating:** 4
**Confidence:** 4

**Summary:**

This work proposes SBMG, a motion generation method that dynamically balances textual and trajectory signals based on their temporal information density. It significantly improves semantic alignment and motion control accuracy in multimodal motion generation.

**Strengths:**

1.It explains the relationship between signal density and control effectiveness.
2.The proposed method achieves state-of-the-art performance.

**Weaknesses:**

Major issues:

1.What exactly are static or simple fusion strategies? Please provide an explanation.
2.Provide a quantitative example of trajectory deviation or semantic drift caused when one signal dominates. Conduct both quantitative and qualitative analyses of the relationship between signal density and control effectiveness to justify the value of this contribution.
3.In Equation 1, the paper defines it as a score computation, but after multiplying the softmax-normalized attention scores with X, it becomes a feature rather than the intended score. Therefore, the definitions of dense and sparse semantic scenarios are questionable.
4.Why is the TSB module not mentioned in the abstract or introduction? Does TSB contribute to behavior improvement? Please include relevant experiments.
5.What are the limitations of the proposed method? When textual descriptions become complex, does the information density theory still hold?

Minor issues:

1.In Figure 3, do the different colored text boxes represent different tokens, and how are they related to semantic signal density?

2.Please bold vectors in equations, use commas “,” in the middle of equations, and periods “.” at the end.

3.What is SDM, and why can it extract information density?

4.Why are the drawn framework diagrams not cited?

**Questions:**

See Weakness.

---

### Official Review · Reviewer_rGUn · 2025-10-31

**Soundness:** 2
**Presentation:** 3
**Contribution:** 3
**Rating:** 4
**Confidence:** 4

**Summary:**

The paper argues that prior work mostly fuses or averages trajectory and textual signations when injecting them into the model, which leads to conflicts like following the text but drifting off the trajectory or tracking the path but losing semantics. The authors propose SBMG, which has a Signal Density Measuring (SDM) module that at each timestep t estimates how dense the conditioning information is, and also a Text–Trajectory Signal Balancing (TSB) module that uses those dense/sparse signals to generate a gating factor to modulate both conditioning information. The proposed module can be easily plugged into multiple backbones (OmniControl, MoMask, SALAD). And experimental results indicate better performance in metrics like FID and R-Precision on standard motion generation benchmarks.

**Strengths:**

1. The perspective to rethink about condition injection is quite interesting. Though previous works such as SALAD, MARDM, CLOSD, and ACMDM also test different conditioning injection methods (from in-context to cross attention or AdaLN) that significantly enhance, I believe this is the first work to rethink in an information density way.

2. The controlled experiment results show great potential for this method, as using the same baseline, using SBMG consistently outperforms their baseline methods.

[SALAD]Hong, Seokhyeon, et al. "SALAD: Skeleton-aware Latent Diffusion for Text-driven Motion Generation and Editing." CVPR 2025

[CLoSD]Tevet, Guy, et al. "CLoSD: Closing the Loop between Simulation and Diffusion for multi-task character control." ICLR 2025.

[MARDM]Meng, Zichong, et al. "Rethinking Diffusion for Text-Driven Human Motion Generation." CVPR 2025.

[ACMDM]Meng, Zichong, et al. "Absolute Coordinates Make Motion Generation Easy." ArXiv 2025

**Weaknesses:**

1. Many works (MARDM, MotionStreamer) have questioned the validity of the original evaluation method from HumanML3D, as the method significantly surpasses ground truth, while the visual quality does not align with the evaluation results. This is a valid concern, as methods such as the proposed one in the paper and SALAD are way better than the ground truth itself, but SALAD's visualization still exhibits noticeable jittering shown on their website. Would the author follow new evaluation methods for fairer comparison?

2. Though noted as optional by the conference, video results are an important part of motion generation methods. It is quite hard to tell if the method improves qualitatively without any video results.

3. There are many new methods after OmniControl that enhance motion control tasks, such as MotionLCMV2, MaskedControl, and ACMDM. Some of these methods show flawless control with original injection methods. I believe including a comparison with newer methods would make the experiment more complete.

[MotionStreamer]Xiao, Lixing, et al. "MotionStreamer: Streaming Motion Generation via Diffusion-based Autoregressive Model in Causal Latent Space." ICCV 2025.

[MotionLCMV2]Dai, Wenxun, et al. "Real-time Controllable Motion Generation via Latent Consistency Model." ArXiv 2024

[MaskControl]Pinyoanuntapong, Ekkasit, et al. "Controlmm: Controllable masked motion generation." ICCV 2025

**Questions:**

1. I am curious for an explanation from the author on SALAD(ours), where R-Precision is significantly higher than ground truth itself, but FID is much worse. Intuitively, this means the generated motions are much better following text, but are not the same as the ground truth motion. This counters common sense in a way, since better text following should usually mean similar to real motions, as HumanML3D data are real-human-collected data.

---

### Official Review · Reviewer_bjoX · 2025-11-01

**Soundness:** 3
**Presentation:** 3
**Contribution:** 3
**Rating:** 4
**Confidence:** 4

**Summary:**

This paper proposes Signal Density Measuring (SDM) to detect dense/sparse semantic periods from cross-attention and a Time-Series Balancing (TSB) block that dynamically trades off text vs. trajectory through time. The proposed modules are plug-and-play and are inserted into MoMask, SALAD, and OmniControl with reported gains on HumanML3D/KIT-ML.

**Strengths:**

1. Simple, backbone-agnostic plug-and-play modules that can be applied to MoMask/SALAD/OmniControl without architecture changes.

2. The motivation of the paper and the method are intuitive and clear, that is to balance text and trajectory.

3. Good quantitative results that improve baseline models.

**Weaknesses:**

The major concern is for the experiment.

1. No demo videos in the supplementary file. Quantitative metrics in text-to-motion is proven to be fragile and sometimes misaligned with human judgment. For motion, demo videos are necessary. I don’t see any supplementary videos, which makes it hard to judge the actual quality. The R-Precision being even higher than the ground-truth is meaningless and cannot reflect visual quality.

2. Missing strong baseline results. Compare against recent, stronger works (MARDM [1], MotionStreamer [2], MotionLCM v2 [3]) for the text-to-motion generation task, and MaskControl [4] for the control task, with both qualitative and quantitative results.

3. The main contribution of this paper is balancing text and trajectory, yet the main Table 1 for MoMask/SALAD reports only text-based metrics. I'm a bit confused why balancing text and trajectory can improve the T2M generation itself (If I'm wrong the author can correct me on this point)

4. From the qualitative evaluation in the appendix, I can’t see any clear difference between MoMask and Ours. I feel like the MoMask results are quite good. This echoes point 1 that video results are required.

References:

[1] Rethinking Diffusion for Text-Driven Human Motion Generation

[2] MotionStreamer: Streaming Motion Generation via Diffusion-based Autoregressive Model in Causal Latent Space

[3] MotionLCM-V2: Improved Compression Rate for Multi-Latent-Token Diffusion

[4] MaskControl: Spatio-Temporal Control for Masked Motion Synthesis

**Questions:**

Refer to Weaknesses.

---

### Official Review · Reviewer_7QtU · 2025-11-01

**Soundness:** 2
**Presentation:** 1
**Contribution:** 1
**Rating:** 2
**Confidence:** 5

**Summary:**

This paper introduces an innovative framework for human motion generative with both textual description and trajectory control. These two signals are processed with balaced information density to improve the generation quality.

**Strengths:**

1. The paper frames the relationship between text and trajectory as 'text-primary with trajectory as a generation-time control signal' which is conceptually clean and potentially general.

**Weaknesses:**

1. Introduction issues. The works cited at L36 are not methods that use only text as input; L39 should cite relevant prior work but currently does not. PriorMDM is not mentioned at L50, yet it is later singled out for targeted analysis. In addition, the discussion is also not rigorous. For example, it omits the fine-tuning stage of PriorMDM and overlooks more recent developments such as InterControl's gradient-based correction. Overall, the exposition is loose and unconvincing.

2. The quantitative claims in the abstract (e.g., 60.5% FID reduction) do not reconcile with Table 1 under MoMask/SALAD backbones; please provide exact computations and per-seed raw results, or revise the claim accordingly.

3. The main text centers on MoMask, with cross-backbone evidence limited; moving OmniControl to the appendix reinforces the impression that the approach is MoMask-specific. Qualitative examples also focus mainly on MoMask. The main motivation is the balancing of textual control and spatial control. Hence, it is strange to use MoMask as the main baseline network. In addition, the authors should clarify more on how the proposed method can help the method with sole textual input.

4. More qualitative results are needed. Currently, there are only four examples in the appendix and only on MoMask (with does not receive sptial signal). The authors should supply more examples on OmniControl and provide comprehensive user study to support their claims.

**Questions:**

1. In table 6, is it correct that the performance of OmniControl on "Left Hand" setting is exactly same as that on "Right Hand" setting?

---

### Note · Authors · 2025-11-12

I have read and agree with the venue's withdrawal policy on behalf of myself and my co-authors.